# Growth State-Dependent Expression of Arachidonate Lipoxygenases in the Human Endothelial Cell Line EA.hy926

**DOI:** 10.3390/cells11162478

**Published:** 2022-08-10

**Authors:** Mohammad G. Sabbir, Jeffrey T. Wigle, Carla G. Taylor, Peter Zahradka

**Affiliations:** 1Canadian Centre for Agri-Food Research in Health and Medicine, St. Boniface Hospital Albrechtsen Research Centre, Winnipeg, MB R2H 2A6, Canada; 2Institute of Cardiovascular Sciences, St. Boniface Hospital Albrechtsen Research Centre, Winnipeg, MB R2H 2A6, Canada; 3Department of Biochemistry and Medical Genetics, University of Manitoba, Winnipeg, MB R3E 0J9, Canada; 4Department of Food and Human Nutritional Sciences, University of Manitoba, Winnipeg, MB R3T 2N2, Canada; 5Department of Physiology and Pathophysiology, University of Manitoba, Winnipeg, MB R3E 0J9, Canada

**Keywords:** endothelial cells, lipoxygenase, phenotype, bioactive lipids

## Abstract

Endothelial cells regulate vascular homeostasis through the secretion of various paracrine molecules, including bioactive lipids, but little is known regarding the enzymes responsible for generating these lipids under either physiological or pathophysiological conditions. Arachidonate lipoxygenase (ALOX) expression was therefore investigated in confluent and nonconfluent EA.h926 endothelial cells, which represent the normal quiescent and proliferative states, respectively. mRNAs for *ALOX15*, *ALOX15B*, and *ALOXE3* were detected in EA.hy926 cells, with the highest levels present in confluent cells compared to nonconfluent cells. In contrast, *ALOX5*, *ALOX12*, and *ALOX12B* mRNAs were not detected. At the protein level, only ALOX15B and ALOXE3 were detected but only in confluent cells. ALOXE3 was also observed in confluent human umbilical artery endothelial cells (HUAEC), indicating that its expression, although previously unreported, may be a general feature of endothelial cells. Exposure to laminar flow further increased ALOXE3 levels in EA.hy926 cells and HUAECs. The evidence obtained in this study indicates that proliferative status and shear stress are both important factors that mediate endothelial *ALOX* gene expression. The presence of ALOX15B and ALOXE3 exclusively in quiescent human endothelial cells suggests their activity likely contributes to the maintenance of a healthy endothelium.

## 1. Introduction

The endothelial cells lining the inner surface of blood vessels are responsible for regulating vascular homeostasis [1], which includes modulation of vascular tone and blood flow, movement of nutrients and waste products between the blood and the underlying tissues, thrombogenicity, hemostasis, and inflammation [2]. To maintain this homeostatic balance, endothelial cells must be able to rapidly respond to various physiological signals. Endothelial cells are also major players in the processes that are activated when vascular tissue is damaged by pathophysiological stressors, such as oxidant damage, vascular surgical procedures, hyperlipidemia, hypertension, and disturbed blood flow [3,4]. Under these conditions, endothelial cells respond by modulating their underlying properties to enable migration, proliferation, and extracellular matrix remodeling [5]. This alteration of growth state, from quiescent to proliferating, which has been termed activation or endothelial to mesenchymal transition (EndMT), is essential for efficient repair of injured vascular tissue [6,7]. Growth state also plays a role in the response to changes in blood flow and remodeling of the blood vessel [8,9]. Endothelial cell activation results in various physical changes to the cell that affect its adhesion and barrier properties, as well as alterations in the production of paracrine molecules responsible for mediating the inflammatory response and recruitment of immune cells [10,11].

The paracrine functions of endothelial cells are primarily associated with control of vascular tone, however, they also are closely affiliated with inflammation [12]. For these reasons, the production and secretion of endocrine factors is regulated as a function of the endothelial growth state [13]. While considerable focus has been placed on nitric oxide (NO), since suppression of its production during endothelial cell activation promotes efficient tissue repair by enabling cell proliferation and cellular adhesion [14], the activation state also influences the secretion of numerous other paracrine/autocrine agents [15]. Furthermore, in addition to modulating tone, these agents, particularly those classified as bioactive lipid mediators, are required for regulating the progression and the eventual resolution of tissue repair as it relates to inflammation [16,17]. With respect to vascular tissues, the profile of released bioactive lipid mediators is different for each cell type; but, at this point in time, significantly more is known regarding those lipid mediators produced by circulating immune cells, such as platelets and tissue macrophages rather than endothelial cells [18,19].

The bioactive lipids most closely linked with vascular function are the eicosanoids, which are synthesized mainly from arachidonic acid (AA), an omega-6 (n-6) polyunsaturated fatty acid (PUFA) [20]. The cyclooxygenase (COX)-derived prostaglandins (PGs) and thromboxanes (TXs), in particular, modulate blood vessel contractility, and contribute to inflammation and vascular homeostasis [21]. Furthermore, both COX-1 and COX-2 are expressed in endothelial cells and their genes are more active in proliferating cells [22]. Likewise, the literature suggests endothelial cells express one or more active arachidonate lipoxygenases (ALOXs) as indicated by the synthesis of hydroxyperoxyeicosatetraenoic acids (HpETEs) and their hydroxyeicosatetranoic acid (HETE) metabolites [23,24], however, it remains unclear which ALOX enzymes are found in endothelial cells and how their expression is regulated.

Lipoxygenases are non-heme iron-containing dioxygenases that catalyze the stereo-specific peroxidation of free and esterified PUFAs to generate a spectrum of bioactive lipid mediators [25]. There are six *ALOX* genes in humans, designated *ALOXE3*, *ALOX5*, *ALOX12*, *ALOX12B*, *ALOX15*, and *ALOX15B* [26]. The nomenclature of ALOXs is based on the carbon position in the PUFAs at which a single oxygen molecule is incorporated [26]. The expression profile for *ALOX* genes depicted in the human protein atlas (HPA) indicates they exhibit organ, tissue, and cell-type-specific expression [27]. With respect to endothelial cells, it has been reported that human endothelial cells of different origins express *ALOX5* [28], *ALOX12* [29], and *ALOX15* [30]. On the other hand, single-cell RNA sequencing-based transcriptional profiling of mouse aorta identified relatively high expression of *ALOX5* in the aortic monocytes and resident macrophages but failed to detect *ALOX* expression in endothelial cells [31]. These conflicting reports indicate a comprehensive examination of *ALOX* expression in endothelial cells is warranted. This study was therefore designed to examine *ALOX* gene expression in human EA.hy926 endothelial cells by measuring the relative mRNA and protein levels of all 6 ALOXs. Their expression profile was also compared in primary Human Umbilical Artery endothelial cells (HUAECs), confluent (quiescent), and non-confluent (proliferating) cells, as well as in cells exposed to flow, to determine whether differences in expression exist between the healthy and the dysfunctional endothelial states.

## 2. Materials and Methods

### 2.1. Cell Culture 

EA.hy926, HEK293, HepG2, MCF7, and MDA MB231 cells, all purchased from the American Type Culture Collection (Manassas, VA, USA), were grown in Dulbecco’s modified Eagle’s medium (DMEM), supplemented with 10% heat-inactivated FBS and 1× antibiotic-antimycotic solution (A5955; Sigma-Aldrich Canada, Oakville, ON, Canada). EA.hy926 cells were cultured for 7–20 days with media changed every second day to achieve 100% confluent cells [32]. Non-confluent (10–20%) cells were obtained 24 h after 1:4 subculture of a confluent plate of cells. THP-1 cells were grown in RPMI-1640 medium supplemented with 0.05 mM 2-mercaptoethanol and 10% FBS [33]. Pooled human umbilical artery endothelial cells (HUAECs) were purchased commercially (Catalog number: C-12202; Cedarlane, Burlington, ON, Canada) and grown in Vascular Cell Basal Media (Catalog number: PCS-100-030; ATCC) supplemented with Endothelial Cell Growth Kit-VEGF (Catalog number: PCS-100-041; ATCC). The HUAECs were subcultured at 80% confluency and used within 3–5 passages. 

### 2.2. RNA Extraction, cDNA Synthesis, and Reverse-Transcription PCR (RT-PCR) 

Total RNA isolation, first-strand cDNA synthesis, and RT-PCR were performed in a 20 µL reaction mix as previously described [34]. The RT-PCR products were subsequently visualized after agarose gel electrophoresis. Primer sequences are provided in Table 1.

### 2.3. Plasmids and Transfection 

Full-length C-terminal Flag-tagged ALOX ORFs were purchased from GenScript. Transfection of plasmids into EA.hy926 cells was performed using the Amaxa nucleofection reagent (Catalog number: VP1-1001; Lonza, Cohasset, MN, USA). 

### 2.4. Dicer-Substrate siRNA (DsiRNA) Mediated Knockdown

Three 27 nucleotide siRNA duplexes specific for ALOXE3 (Table 1) and a TYE-563 tagged transfection control were purchased (DsiRNA TriFECTa Kit; Integrated DNA Technologies, Coralville, IA, USA) and reconstituted according to the manufacturer’s protocol. Transfection of the DsiRNAs into EA.hy926 cells was achieved using Lipofectamine MessengerMAX Reagent (ThermoFisher Scientific, Waltham, MA, USA). The transfected cells were incubated for 48 h, at which time brightfield and fluorescent images were captured with an Axiovision 4.1 Zeiss microscope. Cells were subsequently harvested, and ALOXE3 was measured.

### 2.5. Western Blotting

The proteins-of-interest were analyzed by immunoblotting after SDS-PAGE separation of total proteins and transfer to nitrocellulose membrane. Relative quantification was obtained using ImageJ Software (version 1.48) [35], as previously described [36]. Table 1 lists the primary antibodies used in this study. 

### 2.6. Endothelial Cell Perfusion for Simulation of Laminar Flow

Exponentially growing EA.hy926 cells or HUAEC were seeded into the μ-Slide I Luer channel slides (ibidi USA Inc, Fitchburg, WI, USA) with a channel height of 0.4 mm, as per the manufacturer’s instructions. The cell seeding parameters were adjusted so confluency in the μ-Slide I Luer channel slides was reached within 24 h. Subsequently, the μ-Slide I Luer channel slides were individually connected to the perfusion system and attached to the ibidi fluidic unit and pump. The flow rate (φ: mL/min) in the μ-Slide I Luer slides was held at a constant 32 mL/min to achieve a shear force (τ: dyn/cm^2^) of 30 Dyn/cm^2^ based on the formula τ = η × 131.6 × φ, where η = dynamical viscosity (Dyn-s/cm^2^). The viscosity of the cell culture medium with 10% serum was considered to be 0.0072 Dyn-s/cm^2^. The cells were exposed to flow for 24 h while the control cells were maintained under static conditions over the same period. The cells were imaged via brightfield microscopy, then harvested for Western blotting.

### 2.7. Statistical Analysis 

Statistical analysis was performed using Prism version 7.00 (GraphPad Software, San Diego, CA, USA). The means of more than two groups were compared using one-way ANOVA (randomized), followed by Dunnett’s post hoc multiple comparison test to compare the means of multiple experimental groups against a control group mean. Comparisons between two groups were performed using Student’s *t*-test (unpaired). Differences were considered significant at *p* < 0.05. 

## 3. Results

### 3.1. RT-PCR Detects ALOXE3, ALOX15, and ALOX15B mRNAs in EA.hy926 Cells

*ALOX* gene expression by EA.hy926 endothelial cells was profiled using reverse transcription polymerase chain reaction (RT-PCR) with primers (Table 1) designed to specifically amplify mRNA transcribed from the 5 ALOX genes (*ALOXE3*, *ALOX12*, *ALOX12B*, *ALOX15*, *ALOX15B*) clustered on human chromosome 17p13.1, as well as the *ALOX5* gene located on chromosome 10q11.21. RNA was isolated from EA.hy926 cells that were 10–20% and 100% confluent, conditions that we have previously shown represent proliferative and quiescent endothelial cells, respectively [32]. Agarose gel electrophoresis of the RT-PCR products revealed the presence of *ALOXE3*, *ALOX15*, and *ALOX15B* mRNA in 100% confluent EA.hy.926 cells (Figure 1A), while *ALOX5*, *ALOX12*, and *ALOX12B* were not detected. Interestingly, no *ALOX* amplification products were obtained with RNA isolated from the 10–20% confluent EA.hy926 cells (Figure 1B). The absence of *ALOXE3*/*ALOX15*/*15B*-specific amplification products in the non-confluent EA.hy926 cells suggests growth state, possibly linked to cell contact, may modulate *ALOX* expression in this cell type. In comparison, THP-1 cells expressed *ALOX5* and *ALOX15* (Figure 1C), and HepG2 cells expressed *ALOXE3*, *ALOX5*, and *ALOX15* (Figure 2D). THP-1 and HepG2 were chosen as references for comparison with EA.hy926 cells because the Human Protein Atlas (HPA) transcriptome data archive describes positive *ALOX* expression in these cell lines [27]. These data, along with the results presented in our previous study [34], validate the primer pairs designed for assessing expression of these genes, and thus confirm that *ALOXE3*, *ALOX15*, and *ALOX15B* mRNA were only detected in confluent EA.hy926 endothelial cells. The complete *ALOX* gene expression profile for the various cell types examined is summarized in Figure 1E.

### 3.2. Differential Cadherin (CDH5) Expression Distinguishes the Activated (Non-Confluent) and Quiescent (Confluent) EA.hy926 Cell Growth States

To confirm that differences in growth state influence *ALOX* expression in EA.hy926 endothelial cells, immunoblotting for proteins capable of distinguishing the activated versus the quiescent state was performed. Previously, sub-confluent and confluent EA.hy926 cells were characterized in terms of DNA synthesis and the presence of cell division and adherent junction (CDH5)-associated protein [32], an endothelial-specific adhesion molecule located at adherens junctions [37]. Bright-field images (Appendix A) provided visual evidence of significant cell–cell contact in the confluent cell cultures, whereas non-confluent cells showed substantially fewer contacts. The level of CDH5 was subsequently examined to confirm the state of the cell–cell contact, since the abundance CDH5 was previously shown to be lower in sub-confluent cells compared to confluent cells. We reported that the relative abundance of CDH5 gradually increases following initial plating of the cells, reaching a peak after 7 days of culture [32]. In accordance with these observations, CDH5 levels were much higher in confluent versus non-confluent EA.hy926 cells (Appendix A). Likewise, CDH13 (T-Cadherin) was markedly higher in confluent cells (Appendix A). Furthermore, while the total protein profile showed distinct differences between confluent and non-confluent cells (Appendix A) indicative of significant differences in the protein content of these cell states, the total protein levels were similar, thus confirming that CDH5 and CDH13 were significantly reduced in non-confluent cells. Overall, these results verify that *ALOX* mRNA levels correlate with the degree of EA.hy926 cell confluency, thus suggesting cell–cell contact and cell growth state are important factors for modulating *ALOX* gene activity.

### 3.3. ALOX15B mRNA, but Not ALOX15 mRNA, Is Translated in EA.hy926 Cells

Immunoblotting with an anti-ALOX15 antibody was used to examine the protein levels of this enzyme in confluent EA.hy926 cells, the condition under which its mRNA was detected. Surprisingly, the anti-ALOX15 antibody (MA5-25653, Invitrogen) failed to amplify a band in EA.hy926 cells (Figure 2A). Since ALOX15 has been previously reported in endothelial cells [38,39], further investigation of this discrepancy was carried out. 

The HPA transcriptome database was screened for cancer cell lines expressing high-levels of ALOX15 to use as positive controls for ALOX15 protein expression. High *ALOX15* mRNA levels were identified in metastatic breast tumor-derived epithelial-like MCF7 cells [40]. MDA-MB231 cells, which were established from pleural effusions from breast cancer patients [41], were also used for this purpose. Immunoblotting using anti-ALOX15 antibody revealed the presence of two immunoreactive bands in the 63–75 kDa range in lysates from both MCF7 and MDA-MB231 cells (Figure 2B). 

As a positive control, C-terminal Flag-tagged ALOX15 (ALOX15-Flag) cDNA (Transcript ID: ENST00000293761.8) was synthesized and cloned into pcDNA3.1(+). This construct was transiently transfected into HEK293 cells. Anti-Flag antibody immunoblotting revealed the appearance of two bands in overexpressed HEK293 cells, a predominant 74 kDa band and a less abundant 58 kDa band (Figure 2C). These bands were not detected in HEK293 cells transfected with an empty vector. Immunoblotting using the anti-ALOX15 antibody also detected both p74 and p58 ALOX15-Flag proteins in overexpressed HEK293 cells only, indicating that HEK293 cells do not contain native ALOX15. The theoretical molecular weight of ALOX15 (protein ID: ENSP00000293761) was calculated to be 74.8 kDa with the ExPASy Compute pI/MW tool [42]. ALOX15 has an in-frame alternative second AUG start codon that can generate a shorter 57.5 kDa protein (Appendix A). Thus, the presence of the 74 kDa and 58 kDa proteins in ALOX15-Flag overexpressed cells indicates that both start codons may be used during translation, however, the relative band intensities suggest that usage of the alternative second start codon is less than the first start codon. Overall, these results indicate that ALOX15 protein is not produced in confluent EA.hy926 cells. 

Since the vascular localization of ALOX15B is predominantly associated with immune cells [43], the presence of ALOX15B mRNA in EA.hy926 cells was not expected. Immunoblotting was therefore used to determine if ALOX15B protein was also present in these cells. However, due to the similarities between ALOX15 and ALOX15B, the specificity of the anti-ALOX15B antibody was examined. C-terminal Flag-tagged *ALOX15B* (ALOX15B-Flag) cDNA (Transcript ID: ENSP00000369520) was synthesized and cloned into pcDNA3.1(+), then transiently transfected into HEK293 cells. No bands were detected in untransfected HEK293 cells using either anti-FLAG or anti-ALOX15B antibodies, indicating the native protein is not present in these cells (Figure 3A), in agreement with the HPA database. However, the anti-ALOX15B antibody (sc-271290, Santa Cruz Biotechnology) detected 3 bands at 73 kDa, 67 kDa, and 51 kDa (Figure 3A). These bands were distinct from the 58 kDa and 73 kDa bands detected with anti-ALOX15 antibody. Since the ALOX15 and ALOX15B bands were also unique according to the anti-FLAG antibody (Figure 3A), it could be confidently concluded that both proteins can be distinguished by immunoblot analysis.

The 73 kDa ALOX15B band corresponds to the theoretical molecular weight of ALOX15B (72.5 kDa) calculated with the ExPASy Compute pI/MW tool [42]. However, ALOX15B has in-frame alternative second and third AUG start codons that can generate 56.5 and 50.8 kDa proteins, respectively (Appendix A). The presence of p73, p67, and p51 bands in ALOX15-Flag overexpressed cells indicated that all three start codons are used during translation, although it would appear that the second alternative start codon was the most active based on their relative band intensities (Appendix A). Of note, the 57 kDa ALOX15B-Flag appeared at 67 kDa with respect to the molecular weight markers, likely due to the presence of post-translational modifications.

When applied to lysates of EA.hy926 cells, the ALOX15B antibody visualized a single band of 67 kDa (Figure 3B) in samples from confluent cells, which was absent from the non-confluent cell samples. While these results might indicate only one start codon is active in endothelial cells, the weak band intensity suggests the cellular content of ALOX15B is low and the presence of the 73 and 51 kDa bands might be below the limit of detection. However, based on these results, it is reasonable to suggest that the *ALOX15B* mRNA in confluent EA.hy926 cells is translated into protein.

### 3.4. Identification of ALOXE3 Protein in Endothelial Cells

Unlike other ALOXs, ALOXE3 has never been reported in endothelial cells. In order to corroborate the presence of ALOXE3 protein, immunoblotting was selected as the most efficient method. An approach to validating the ALOXE3 antibody similar to that of ALOX15 and ALOX15B was considered, but the ALOXE3 antibody (PA5-21833, Invitrogen) detected 3 bands (70, 100 and 150 kDa) in HEK293 cells (Appendix A), which is not surprising since the HPA database reports ALOXE3 is expressed in these cells. However, since the antibody also recognized a similar trio of bands in EA.hy926 cells (not shown), use of transient transfection for validation was considered unfeasible. Consequently, an mRNA knockdown strategy was employed to establish which band(s) represent ALOXE3, and HEK293 cells were used for this purpose since they are much easier to transfect than EA.hy926 cells.

Dicer-Substrate siRNAs (DsiRNAs), which are chemically synthesized 27mer RNA duplexes, exhibit RNA knockdown potency up to 100-fold more than conventional 21mer siRNAs [44]. Three sets of DsiRNA, each targeting a different exon of the *ALOXE3* gene (Figure 4A), were designed and transfected into HEK293 cells. Immunofluorescence images of the fluorescence-conjugated control DsiRNA (Dsi NC) revealed high transfection efficiency was achieved using lipofectamine (Figure 4B). Immunoblotting for ALOXE3 revealed knock down by the three DsiRNAs only affected the 100 kDa band (Figure 4C), while the levels of the 70 and 150 kDa bands remained constant. These results indicate that the 70 and 150 kDa immunoreactive bands are non-specific and that the 100 kDa immunoreactive band represents the ALOXE3 protein. Quantification of the 100 kDa band intensities after normalization to GAPDH revealed that DsiRNA3 was the most effective in knocking down ALOXE3 (Figure 4D). Interestingly, the identification of the 100 kDa band as ALOXE3 (protein ID: ENSP00000314879) suggests it may be post-translationally modified since the apparent molecular mass is higher than the predicted size of 80 kDa, based on calculations with the ExPASy Compute pI/MW tool [42]. However, use of an in-frame alternative second AUG start codon may cause translation of a 76.2 kDa (theoretical) ALOXE3 protein (Appendix A), even though transcriptional isoforms are not listed in the Ensembl human genome assembly (GRCh38.p13) database [45].

Immunoblotting for ALOXE3 detected the three immunoreactive bands (70, 100, and 150 kDa) in confluent EA.hy926 cells observed in HEK293 cells but failed to detect the 100 kDa band representative of ALOXE3 in non-confluent EA.hy926 cells (Figure 5A). These results correlated with the observed differences in the levels of *ALOXE3* mRNA under these same growth conditions (Figure 1A,B). Based on these observations, it was concluded that *ALOXE3* is only expressed in quiescent EA.hy926 endothelial cells.

To verify that the results obtained with EA.hy926 cells were representative of endothelial cells, ALOXE3 protein levels were also measured in primary human endothelial cells derived from umbilical cord arteries (HUAEC). As was seen with EA.hy926 cells, immunoblotting detected ALOXE3 protein in confluent HUAEC cells (Figure 5A). Furthermore, the relative abundance of ALOXE3 in HUAEC cells was similar to that of EA.hy926 cells when normalized relative to total protein (Figure 5B). Thus, *ALOXE3* is likely normally expressed in quiescent endothelial cells but not when they are in the activated state.

### 3.5. Shear Force Influences the Levels of ALOXE3 Protein in Endothelial Cells

In addition to cell contact, endothelial cell growth and activation can be influenced by blood flow. Specifically, shear stress can induce endothelial cell quiescence much like cell contact [46]. To see if *ALOXE3* expression can be modulated by shear stress, confluent EA.hy926 were exposed to shear forces of 0, 8, 14, and 30 Dyn/cm^2^ for 24 h, which approximate the physiological forces observed in veins, ascending aorta, and arteries, respectively [9]. Brightfield microscopy revealed that application of shear force for 24 h led to a loss of confluency and formation of isolated patches of 4–20 cells (Figure 6A). This occurred under all conditions, however, only 30 Dyn/cm^2^ is shown. In contrast, HUAECs exposed to 30 Dyn/cm^2^ (lower shear forces were not tested) remained confluent (Figure 6A). These results suggest that there is a cell-type-specific response with respect to cell–cell contact and adherence under shear stress. This difference may also be due to the transformed somatic hybrid nature (lung epithelial and endothelial) of EA.hy926 cells compared to the primary HUAEC. However, immunoblotting revealed that ALOXE3 levels increased in response to shear force in both EA.hy926 cells and HUAEC, regardless of the force applied (Figure 6B). Quantification of the immunoblots revealed ALOXE3 was significantly increased by 3.2-fold under shear stress conditions (Figure 6C). These novel findings indicate that the abundance of ALOXE3 protein in endothelial cells is responsive to both shear stress and cell growth state.

## 4. Discussion

This is the first study to comprehensively profile *ALOX* gene expression in endothelial cells. The evidence indicates that the *ALOX15*, *ALOX15B*, and *ALOXE3* genes, and no others, are transcribed in EA.hy926 cells. However, at the protein level, only ALOX15B and ALOXE3 are detected. Furthermore, ALOX15B and ALOXE3 are found primarily in confluent endothelial cells relative to non-confluent cells. Since ALOXE3 has not been previously reported in vascular tissues, its presence in HUAEC was examined as confirmation that this protein is also found in primary endothelial cells. These data are the first to imply that ALOXE3 has a function in quiescent endothelial cells, which is the state these cells normally occupy in healthy blood vessels. However, the fact that ALOXE3 levels are increased in both EA.hy926 cells and HUAEC after exposure to flow suggests cell–cell contact and shear stress may independently regulate this gene. Finally, it is unlikely that the ALOX-derived bioactive lipids present in the circulation are a result of endothelial cell activity since the major products detected in plasma and serum originate via ALOX5 and ALOX15 [47,48], which are not detectable at the protein level in either growing or confluent endothelial cell populations.

The biological function of the ALOX enzymes is to synthesize a variety of bioactive lipids, primarily from PUFA, that modulate the response of many different cells and tissues. For this reason, production of these lipid mediators is sensitive to both the species of PUFA available as a substrate and the amount of enzyme in a cell. With respect to endothelial cells, ALOX-derived bioactive lipids can influence many functions, including cell proliferation, endothelial-monocyte interactions, barrier function, cytokine production, and vasoreactivity [49,50]. While their effects on endothelial cells have been well documented, it still remains uncertain which cells are the source of these bioactive lipids. To date, localization of ALOX within the vasculature has been examined indirectly, with conclusions usually based on a reported association between a given lipid species and the ALOX presumed to be responsible for its synthesis [38,51,52]. However, while this approach is often employed, an individual ALOX produces multiple products including those that have been linked to other ALOX isoforms [43]. Likewise, the application of inhibitors for the same purpose [23] is limited due to uncertainties regarding inhibitor specificity [53]. At the same time, direct examination of *ALOX* expression by endothelial cells, whether by antibody staining for protein or PCR amplification for mRNA, has been sparse [39,54]. Furthermore, the cell culture conditions employed vary between the published studies, which makes meaningful comparisons difficult. Thus, more specific information is essential if an understanding is to be achieved regarding the endothelial origin of mediators that contribute to vascular function.

A major strength of the current study is its use of cells that occupy two well characterized growth states that have physiological relevance [32]: subconfluent endothelial cells that are actively growing, and confluent endothelial cells that are quiescent. In agreement with our previous report [32], confluent cells exhibit higher levels of endothelial-specific markers, resulting in a protein profile characteristic of the endothelial phenotype (see Appendix A). In comparison, growing cells express mesenchymal markers indicating they have undergone partial EndMT and are thus in the activated state [55]. Given this ability to manipulate the cell characteristics by eliciting a transition between quiescent and activated endothelial cells, it has been possible to establish that expression of *ALOX15B* and *ALOXE3* is sensitive to phenotype switching. This is the first indication that either ALOX might have a role in endothelial function, although RNA for both *ALOX15B* and *ALOXE3* was identified in endothelial cells of a limited number of tissues by the HPA database. At the same time, the absence of the other ALOXs from endothelial cells raises questions regarding their contribution to the endothelial function.

While the ALOX15 protein was not detected in either growth state, in agreement with our finding that endothelial cells are not sensitive to the effects of the ALOX15 inhibitor PD146176 [56], the fact that its mRNA is present in quiescent endothelial cells suggests there may be conditions that will lead to its translation. Pertinent to this study, the earliest reports of endothelial expression of *ALOX15* included IL-4 in the culture medium [54], and it was subsequently shown that IL-4 induces *ALOX15* expression in confluent cells [39]. It may thus be speculated, based on the proliferation-inducing effects of ALOX15 and its metabolites [57], that the presence of *ALOX15* mRNA in confluent cells provides a means of rapid endothelial cell activation in response to vascular damage without the need to wait for induction of its gene.

Unlike *ALOX15*, which is broadly expressed, *ALOX15B* mRNA and protein appear to be restricted to a small number of tissues [27]. Interestingly, even though ALOX15B and ALOX15 generate the same products, albeit in different proportions [58], the evidence indicates that ALOX15B suppresses cell growth [59], particularly as it relates to the prostate [60,61]. However, to date, the vascular presence of ALOX15B has largely been linked to macrophages [62], which would explain its localization in atherosclerotic lesions [63,64]. It is also possible that endothelial ALOX15B may have been misidentified as ALOX15 since the discovery of ALOX15B is more recent and antibody specificity would not have been validated. Regardless, it is now possible to state that *ALOX15B* appears to be regulated by growth state, however, further investigation to determine whether there is an inverse relationship between ALOX15B and ALOX15 as suggested by these results will require a more sensitive detection method.

The major outcome of this investigation is the discovery that endothelial cells express *ALOXE3*, an isoform first identified in epithelial cells of the skin [25,65,66]. However, unlike the other ALOX enzymes, PUFAs are not substrates for ALOXE3. Rather, ALOXE3, originally identified as hepoxilin synthase, catalyzes the conversion of ALOX12 and ALOX15 products into hepoxilins [66,67], which are then rapidly metabolized into trioxilins. Since hepoxilins have been shown to affect multiple cellular functions analogous to those active in the endothelium [68], ALOXE3 may contribute to a variety of endothelial processes, including modulation of tight junctions. For instance, products of ALOX15 (15-HETE) and ALOX12 (12-HETE) disrupt endothelial tight junctions [49,69] and increase vascular permeability [70]. In the skin, production of hepoxilin (HXA3) by ALOXE3 from the ALOX12 product 12-HpETE is essential for epidermal barrier formation [66]. ALOXE3 may have a similar role in endothelial cells, which would be to divert metabolism of 15-HpETE and 12-HpETE from barrier-disrupting 15-HETE and 12-HETE to hepoxilins that help preserve endothelium integrity. Based on our results, ALOXE3 could work in tandem with ALOX15B to control endothelial barrier function, as it does with ALOX12 in epithelial cells [65,71], since the available evidence indicates aortic lysates produce hepoxilins only when ALOX12 is included in the reaction system [72].

While the link between ALOXE3 and other endothelial functions has not been investigated directly, it has been established that hepoxilins and/or trioxilins induce vasodilation, prevent platelet aggregation, suppress endothelial-monocyte adhesion, inhibit fibrosis, and suppress inflammation [73,74,75]. In contrast, ALOX12 and ALOX15 products (12-HETE and 15-HETE, respectively) enhance monocyte adhesion [76,77], cause vasoconstriction, and induce inflammation [78]. These opposing actions of 12-/15-HETE and the hepoxilins enable us to extrapolate that ALOXE3 is essential for the lipid mediator class switch that occurs during initial inflammatory and terminal resolution stages of the vascular response to injury [11], wherein endothelial cell phenotype shifts from its quiescent state to the active state and back again. It is likely that this dichotomy also explains why the contribution of ALOX12 to vascular function can be different from the effects of 12-HETE treatment [79], as well as explain why ALOX12 has been positively linked to sex-dependent cardioprotection [47].

An examination of the intracellular pathways that regulate ALOX15 and ALOXE3 levels in endothelial cells has not been included in this study. In large part, this deficiency is due to the lack of information currently available regarding lipoxygenase expression, and in the absence of a credible target such an investigation would be impractical. However, based on the findings of this study, some speculation on this topic is possible. Shear stress as a result of laminar flow is a critical factor in maintaining the endothelial state. Specific mechanosensitive ion channels, such as PIEZO1, convey the mechanical effects of flow to a CaMKII-ERK5 signal transduction cascade that modulates expression of the endothelial-specific transcription factors KLF-2 and KLF-4 [80]. Furthermore, recent evidence suggests that ERK5 operates by phosphorylating MEF2, the transcription factor that controls KLF-2 and KLF-4 gene activity [81]. While a role for KLF-2 and KLF-4 in ALOX15B or ALOXE3 gene expression has not yet been identified, Segre et al. [82] have reported that KLF-4 governs the epidermal barrier function, a process which also involves ALOXE3 [63]. Conversely, disturbed flow leads to a reduction in KLF-2 and KLF-4 and conversion to the activated state [83], where ALOX15B and ALOXE3 are lacking (Figure 7). Consequently, as described above, 12-HpETE and 15-HpETE would preferentially be metabolized to 12-HETE and 15-HETE, respectively, which disrupt barrier function via PKCε-mediated phosphorylation of ZO-1 [69]. The loss of barrier function is considered to be the first step in EndMT, which is essential for vascular repair, but it can also result in endothelial dysfunction and atherosclerosis [55]. It is likely that modes of injury other than disturbed flow, including mechanical damage, also lead to EndMT, although they would engage via a distinct set of signal transduction pathways [84]. Once repair is complete, other lipid metabolizing enzymes contribute to the resolution phase of vascular wound healing [11], and these molecules likely promote restoration of the quiescent endothelial state and re-expression of KLF-2 and KLF-4.

ALOXs produce many different bioactive lipid mediators [85] and while the individual products exhibit distinct biological effects on endothelial cells [50] their aggregate effect is difficult to determine since it would be a function of mediator potency, relative concentrations, and the physiological state of the cell (presence/absence of specific receptors and intracellular signaling mediators). However, while there is a good understanding of how bioactive lipid mediators derived from PUFA contribute to the onset and resolution of inflammation in response to injury, [86] little is known of the mechanisms that regulate the ALOX enzymes responsible for generating these lipid mediators.

## 5. Conclusions

The ability to model distinct endothelial cell phenotypes has provided a unique perspective regarding the contributions of ALOXs to the vasculature in both the healthy (quiescent) and activated (growing) states. Furthermore, it is evident that the distinct properties exhibited under these cell culture conditions reflect to a reasonable degree the physiology of the normal and damaged endothelium and that it is also possible to emulate the effect of flow. Altogether, the information gained through this study has made it possible to propose a new paradigm for the physiological role of ALOXs in endothelial cells, with ALOXE3 contributing to phenotype switching and thus influencing multiple endothelial cell functions through its ability to alter the cellular response to other ALOX products. Finally, establishing that ALOX15B and ALOXE3 are only expressed in quiescent endothelial cells or endothelial cells exposed to flow suggests that these enzymes may be suitable targets for therapeutic interventions intended to manage vascular inflammation. This would provide a unique approach for averting endothelial dysfunction and thus interrupting the progression to ischemic heart disease.

## Figures and Tables

**Figure 1 cells-11-02478-f001:**
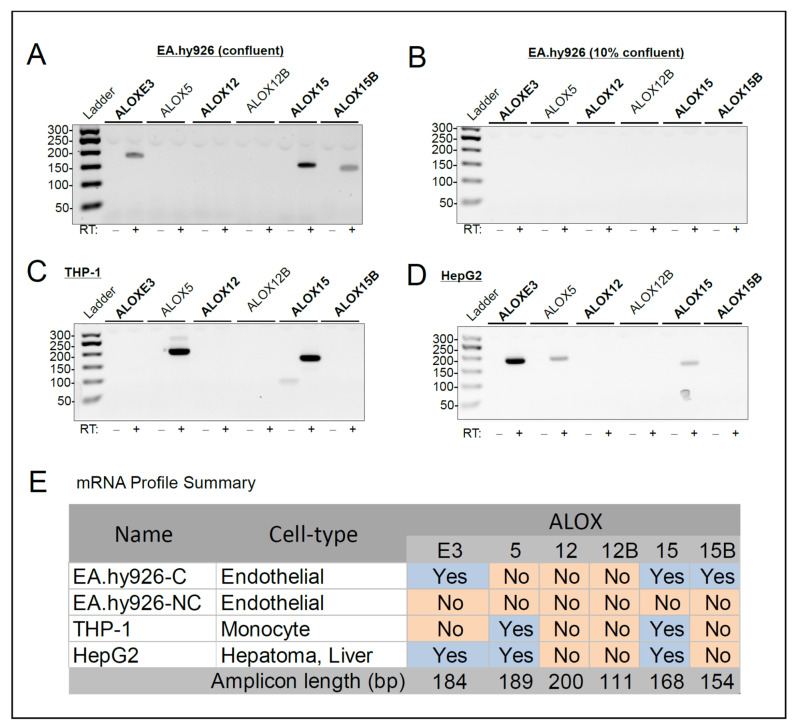
*ALOX* mRNA profile of confluent EA.hy926, non-confluent EA.hy926, THP-1, and HepG2 cells. (**A**–**D**): Agarose gel images showing *ALOX*-specific RT-PCR products for confluent EA.hy926 cells (**A**), non-confluent EA.hy9256 cells (**B**), THP-1 cells (**C**) and HepG2 cells (**D**). RT, reverse transcriptase. (**E**): Table summarizing the ALOX expression profile in the different cell types. C, confluent; NC, non-confluent. Blue colour-coding indicates the presence of *ALOX* mRNA; orange colour-coding indicates *ALOX* mRNA is absent.

**Figure 2 cells-11-02478-f002:**
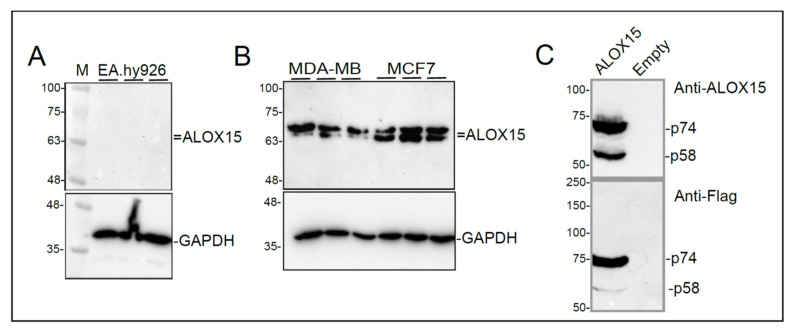
Immunoblot analysis of ALOX15. (**A**): Immunoblots of ALOX15 and GAPDH (loading control) in confluent EA.hy926 cells. Three replicates are shown. M, molecular mass markers. (**B**): Immunoblots of ALOX15 and GAPDH (loading control) in MDA MB231 (MDA-MB) and MCF7 cells. Three replicates are shown for each cell-type. (**C**): Immunoblots of ALOX15 (upper blot) and Flag (lower blot) in HEK293 transiently transfected with Flag-tagged *ALOX15* expression plasmid (ALOX15) or empty plasmid (Empty), as indicated. Molecular mass markers were used to determine the relative size of the bands. Data are representative of three separate experiments.

**Figure 3 cells-11-02478-f003:**
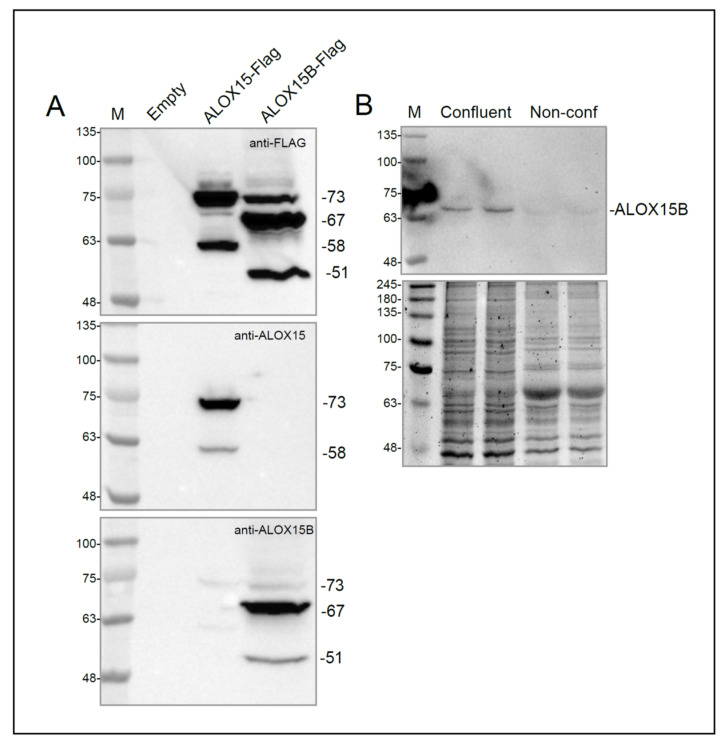
Immunoblot analysis of ALOX15B. (**A**): Immunoblotting of HEK293 cells transiently transfected with control expression plasmid (Empty), Flagged-tagged *ALOX15* expression plasmid or Flag-tagged ALOX15B expression plasmid. Antibodies employed were anti-Flag (upper panel), anti-ALOX15 (middle panel) and anti-ALOX15B (bottom panel). Representative blots are shown. Bands are identified by their molecular mass based on the markers (M). (**B**): Immunoblotting of lysates prepared from confluent or non-confluent EA.hy926 cells with anti-ALOX15B antibody (upper panel). Protein loading is shown by Oriole staining (lower panel).

**Figure 4 cells-11-02478-f004:**
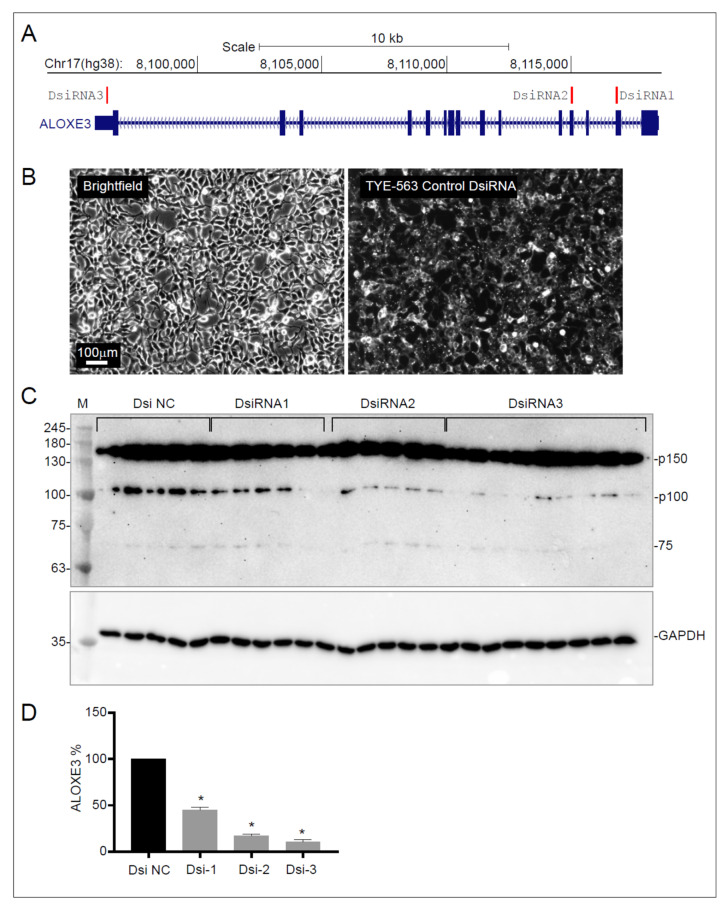
DsiRNA-mediated knockdown of *ALOXE3* in EA.hy926 cells. (**A**): Genomic organization of *ALOXE3* on chromosome 17 showing the location of three DsiRNAs designed for *ALOXE3* knockdown. Blue rectangles indicate exons and red bars indicate the locations targeted by the DsiRNAs. (**B**): Bright field and fluorescent images showing transfection efficiency of TYE-563 labeled control DsiRNA. (**C**): Immunoblots with anti-ALOXE3 and anti-GAPDH antibodies in EA.hy926 cells transfected with control DsiRNA (Dsi NC) and 3 ALOXE3-specific DsiRNAs. The three bands detected by the antibody are labelled according to their approximate molecular mass based on the standards (M). (**D**): Bar graph showing the level of p100 band (only band to decrease in response to DsiRNA transfection and therefore assumed to be ALOXE3) normalized to GAPDH in DsiRNA-transfected EA.hy926 cells. Data represent mean ± SEM. N = 9, 10–15 replicates from 2 independent experiments. Data were analyzed by one-way ANOVA followed by Dunnett’s multiple comparisons test. *, *p* ≤ 0.05; Dsi NC, knockdown control.

**Figure 5 cells-11-02478-f005:**
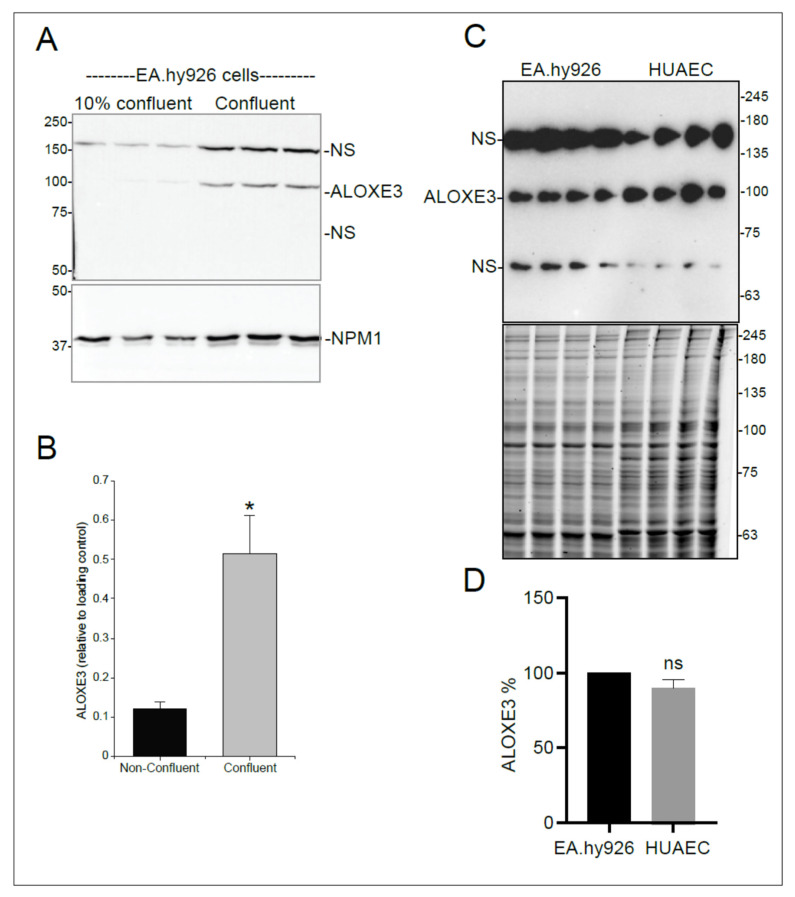
Relative levels of ALOXE3 in confluent and non-confluent EA.hy926 endothelial cells and in confluent HUAEC endothelial cells. (**A**): Lysates of EA.hy926 cells that were either 10–20% confluent or 100% confluent were prepared in triplicate and immunoblotted for ALOXE3 and NPM1 (loading control). Representative blots are depicted in the figure. NS: non-specific bands. (**B**): Bar graph showing the relative levels in confluent and non-confluent EA.hy926 cells of ALOXE3 (100 kDa band) based on densitometry of the bands in panel A normalized to NPM1. Data represent mean ± SEM. N = 3, from 2 independent experiments. Data were analyzed by *t*-test (unpaired). *, *p* ≤ 0.05. (**C**): Immunoblot (top panel) showing levels of ALOXE3 in confluent EA.hy926 compared to confluent HUAEC. Total protein was visualized with Oriole stain (bottom panel) to show relative protein loading. NS: non-specific bands. (**D**): Bar graph showing densitometric quantification of ALOXE3 in EA.hy926 cells and HUAEC after normalization to protein load. Data are presented as mean ± SEM. N = 8 replicates. Data were analyzed by *t*-test (unpaired). ns: not significant (*p* > 0.05).

**Figure 6 cells-11-02478-f006:**
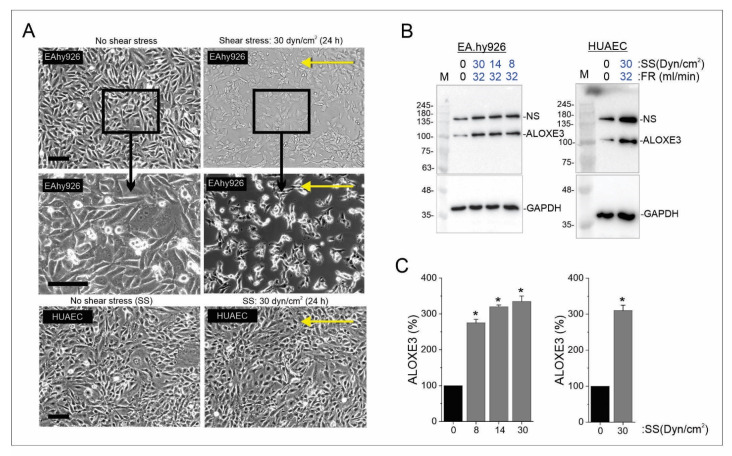
Effect of shear stress on *ALOXE3* expression in EA.hy926 cells and HUAECs. (**A**): Brightfield images showing confluent EA.hy926 cells (top and middle panels) and HUAECs (bottom panels) incubated for 24 h under static (no shear stress) or flow (30 Dyn/cm^2^ shear stress) conditions. Yellow arrows indicate the direction of flow. Middle panels show a magnified view of the region delineated in the upper panels. Scale bar: 100 μm. (**B**): Immunoblots of ALOXE3 and GAPDH (loading control) levels under static (0 Dyn/cm^2^ shear stress) and flow (8, 14, 30 Dyn/cm^2^ shear stress) conditions for EA.hy926 cells and 0 and 30 Dyn/cm^2^ shear stress for HUAECs (passage 3). M: molecular mass markers; NS: non-specific band; SS: shear stress; FR: flow rate. (**C**): Bar graphs showing the levels of ALOXE3 protein normalized to GAPDH based on densitometry of the bands in panel B for EA.hy926 cells and HUAECs. Data are presented as mean ± SEM relative to no shear stress condition (band intensity set to 100%). N = 4 independent experiments. Data were analyzed by unpaired *t*-test. *, *p* ≤ 0.05; SS: shear stress.

**Figure 7 cells-11-02478-f007:**
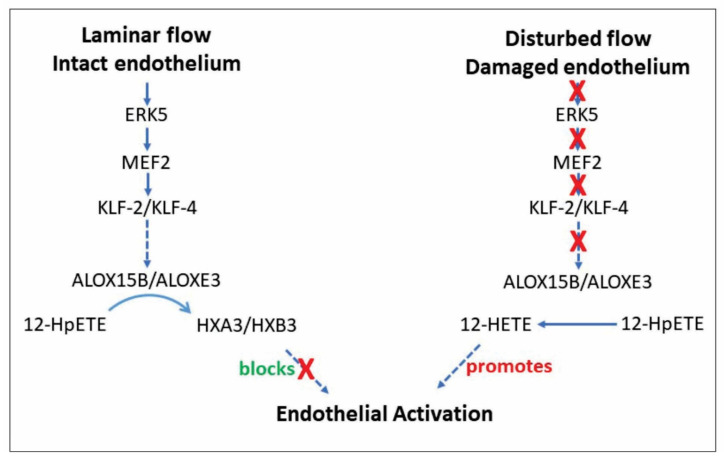
Schematic illustrating possible relationships between the pathways that regulate endothelial cell state in relation to ALOX expression and activity. “Blocks” is presented in green because this action is beneficial, while “promotes” is colored red because this action is associated with the development of vascular disease.

**Table 1 cells-11-02478-t001:** List of primers and antibodies.

Primer Sequences for PCR				
Gene	Forward primer (5′-3′)		Reverse primer (5′-3′)	Amplicon length (base pairs)
ALOXE3	GACGTCAACAGCTTTCAGGAGATG		AGATCGTCTTGGTGGCTGAGTAT	184
ALOX5	GGCAGGAAGACCTGATGTTTGG		GCAGCTCAAAGTCCACGATGAA	189
ALOX12	GAGAAGAGGCTGGACTTTGAATGG		CCATTGAGGAACTGGTAGCTGAAC	200
ALOX12B	GGAAGAGGCTGAAGGACATTAGGA		TGAGGTACTGGTACCCAAAGAAGG	111
ALOX15	CCTTCCTAACCTACAGCTCCTTCT		TCACAGCCACGTCTGTCTTATAGT	168
ALOX15B	GATACACCCTGCACATCAACACA		TCAGGCAGACACAGGAGAGAATAG	154
GAPDH	ACGGGAAGCTTGTCATCAATGG		CCAGTAGAGGCAGGGATGATGT	445
Sequences for DsiRNA				
DsiRNA1	UUCCCCUGCUAUCAGUGGAUUGAAGUGAAGGGGACGAUAGUCACCUAACUUC			
DsiRNA2	CUGCAUGGUAGACGUCAACAGCUTTGGGACGUACCAUCUGCAGUUGUCGAAA			
DsiRNA3	AAGGAAGGAACCGCUUCACUUCUTGCUUUCCUUCCUUGGCGAAGUGAAGAAC			
Antibodies				
Name	Source ^1^	Species	Type	Catalog number
ALOXE3	ThermoFisher	Rabbit	Polyclonal	PA5-21833
ALOX15	ThermoFisher	Mouse	Monoclonal	MA5-25853
ALOX15B	SantaCruz Biotechnology	Mouse	Monoclonal (Clone-D9)	sc-271290
Flag	Sigma-Aldrich	Mouse	monoclonal	F3165
NPM1	SantaCruz Biotechnology	Mouse	Monoclonal (Clone-NA24)	sc-53175
GAPDH	SantaCruz Biotechnology	Mouse	Monoclonal (Clone-D9)	sc-25778

^1^ ThermoFisher Scientific, Waltham MA; SantaCruz Biotechnology, Dallas TX; Sigma-Aldrich Canada, Oakville, ON, Canada.

## Data Availability

Not applicable.

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
