# Peer review of "Growth State-Dependent Expression of Arachidonate Lipoxygenases in the Human Endothelial Cell Line EA.hy926"

_cells, 2022, doi:10.3390/cells11162478_

Round 1
Reviewer 1 Report
In this study, Sabbir et colleagues assessed the expression of arachidonate lipoxygenases (ALOXs) in the human endothelial cells. As suggested by the title, the authors show their exclusive expression in confluent cells. In addition, they show increased ALOXE3 expression under laminar shear stress. The presented data are generally clear and support their conclusion. Addressing the following points will further strengthen the manuscript.
1. The authors propose that ALOXE3 is normally expressed in quiescent/confluent endothelial cells including EA.hy926 cells and HUAECs. However, unlike EA.hy926 cells (Figure 5A), they did not provide data that show ALOXE3 expression in non-confluent HUAECs. They need to add this data to consolidate their argument.
2. In Figure 6, the authors investigated the effect of shear stress on ALOXE3 expression in EA.hy926 cells and HUAECs. They put those cell lines under laminar shear stress of 30 dynes/cm2 for 24 hours. In general, physiological wall shear stress in veins is considered as less than 5-6 dynes/cm2 and that in arteries is considered as 10 to 20 dynes/cm2 although the value varies depending on the size of the vessels. The authors need to explain why they chose 30 dynes/cm2 as their experimental condition. Is there any difference in the expression of ALOXE3 under different extent of laminar shear stress? The authors hypothesized that ALOX expression is regulated by cell-cell contact and cell growth state based on their finding that ALOXE3 was detected in only confluent cells. In Figure 6A, under shear stress for 24 hours EA.hy926 cells are no longer confluent as shown in representative images although ALOXE3 expression was higher in cells under shear stress (non-confluent) compared to cells under the static condition (confluent). It may be useful to do additional experiments with several values of shear stress including a condition that does not induce the loss of cell-cell contact to make sure that both cell proliferative status and shear stress are important mediators of endothelial ALOX gene expression. In addition, as a control experiment, it would be interesting to evaluate the response of cells to a disturbed flow by keeping the cells under oscillatory shear stress.
Author Response
In this study, Sabbir et colleagues assessed the expression of arachidonate lipoxygenases (ALOXs) in the human endothelial cells. As suggested by the title, the authors show their exclusive expression in confluent cells. In addition, they show increased ALOXE3 expression under laminar shear stress. The presented data are generally clear and support their conclusion. Addressing the following points will further strengthen the manuscript.
- The authors propose that ALOXE3 is normally expressed in quiescent/confluent endothelial cells including EA.hy926 cells and HUAECs. However, unlike EA.hy926 cells (Figure 5A), they did not provide data that show ALOXE3 expression in non-confluent HUAECs. They need to add this data to consolidate their argument.
Response: Unfortunately, we do not have a blot to address the point raised by the reviewer that is of publication quality. It would require at least a month to produce such a figure and we have only been given 5 days to complete the revision. We hope that the lack of this one item will be acceptable to the reviewer.
- In Figure 6, the authors investigated the effect of shear stress on ALOXE3 expression in EA.hy926 cells and HUAECs. They put those cell lines under laminar shear stress of 30 dynes/cm2for 24 hours. In general, physiological wall shear stress in veins is considered as less than 5-6 dynes/cm2 and that in arteries is considered as 10 to 20 dynes/cm2 although the value varies depending on the size of the vessels. The authors need to explain why they chose 30 dynes/cm2 as their experimental condition. Is there any difference in the expression of ALOXE3 under different extent of laminar shear stress? The authors hypothesized that ALOX expression is regulated by cell-cell contact and cell growth state based on their finding that ALOXE3 was detected in only confluent cells. In Figure 6A, under shear stress for 24 hours EA.hy926 cells are no longer confluent as shown in representative images although ALOXE3 expression was higher in cells under shear stress (non-confluent) compared to cells under the static condition (confluent). It may be useful to do additional experiments with several values of shear stress including a condition that does not induce the loss of cell-cell contact to make sure that both cell proliferative status and shear stress are important mediators of endothelial ALOX gene expression. In addition, as a control experiment, it would be interesting to evaluate the response of cells to a disturbed flow by keeping the cells under oscillatory shear stress.
Response: We agree with the reviewer that examining the effects of different values of shear stress on AloxE3 expression would be instructive. That is why the experiment with EA.hy926 cells was actually performed under 4 different conditions (0, 8, 14, 30 dynes/cm2) rather than the 2 that were originally shown in Figure 6. As well, the corresponding description in the Results has been updated to relate the shear forces to different vascular beds, and these values have been referenced. This modification should address the main part of the reviewer’s comment, including why 30 Dyn/cm2 was employed (it represents the lower end of the forces experienced by arteries, which are reported to range from 30 to 100). What was observed was that ALOXE3 expression was increased at all shear forces above 0 Dyn/cm2. Since cell-cell contact is lost under these same conditions, it would appear that shear force is a critical factor affecting expression, but that it operates in addition to cell proliferation state (since ALOXE3 levels are higher than in quiescent cells vs those that are proliferating). Unfortunately, we do not have the equivalent data for HUAECs at all shear stress conditions, which was why we originally only showed the two forces that were equivalent. The figure has been modified by expanding the range of data for EA.hy926 cells based on what we had already completed. We also agree that examining the effect of disturbed/oscillatory flow on ALOXE3 expression is of great interest based on our findings, however, that will be the topic for a separate study.
Reviewer 2 Report
The manuscript of Mohammad G Sabbir et al. presents molecular and cell biology investigations on the gene and protein expression profiles of arachidonate lipoxygenases (ALOXs) in the healthy (quiescent) and activated (growing) states of endothelial cells.
Main contribution and novelty of the research:
This research is the first study to comprehensively profile ALOX gene expression in endothelial cells.
Using a combination of biochemical and cell assays the authors are showing that ALOX15B and ALOXE3 are only expressed in quiescent endothelial cells or endothelial cells exposed to laminar flow (shear stress). The data presented so far suggest that these enzymes may be potential pharmacological targets for therapeutic interventions intended to manage vascular inflammation.
The main function of ALOX3 is to mediate the synthesis of hepoxilins and/or trioxilins which are known to induce vasodilation, prevent platelet aggregation, suppress endothelial-monocyte adhesion, inhibit fibrosis and suppress inflammation. The main substrates for ALOX3 are the products of ALOX12 and ALOX15 activity, i.e. (12-HETE) and (15-HETE), respectively.
Importantly, the antagonistic physiological activities of ALOX15B and ALOX3, mainly mediated by the final products of their enzymatic activity, (15-HETE) and trioxilins, respectively, would contribute to the control of the endothelial barrier function.
Moreover, opposing actions of 12-/15-HETE and the hepoxilins support a new function for ALOXE3, i.e., an enzyme essential for the lipid mediator class switch that occurs during initial inflammatory and terminal resolution stages of the vascular response to injury.
Limits of the research and proposed minor revisions:
The paper will benefit from a cartoon schematically showing the possible signaling pathways which triggers the activation of APOX3 in the endothelial cells and comparison with the pathways that regulate APOX15B.
Please add one extra paragraph in the discussion in which you contrast the signal transduction pathways leading to the activation/inhibition of both APOX15B and APOX3 in the endothelial cells under healthy and pathophysiological conditions (like damaged endothelium!).
Author Response
The manuscript of Mohammad G Sabbir et al. presents molecular and cell biology investigations on the gene and protein expression profiles of arachidonate lipoxygenases (ALOXs) in the healthy (quiescent) and activated (growing) states of endothelial cells.
Main contribution and novelty of the research:
This research is the first study to comprehensively profile ALOX gene expression in endothelial cells.
Using a combination of biochemical and cell assays the authors are showing that ALOX15B and ALOXE3 are only expressed in quiescent endothelial cells or endothelial cells exposed to laminar flow (shear stress). The data presented so far suggest that these enzymes may be potential pharmacological targets for therapeutic interventions intended to manage vascular inflammation.
The main function of ALOX3 is to mediate the synthesis of hepoxilins and/or trioxilins which are known to induce vasodilation, prevent platelet aggregation, suppress endothelial-monocyte adhesion, inhibit fibrosis and suppress inflammation. The main substrates for ALOX3 are the products of ALOX12 and ALOX15 activity, i.e. (12-HETE) and (15-HETE), respectively.
Importantly, the antagonistic physiological activities of ALOX15B and ALOX3, mainly mediated by the final products of their enzymatic activity, (15-HETE) and trioxilins, respectively, would contribute to the control of the endothelial barrier function.
Moreover, opposing actions of 12-/15-HETE and the hepoxilins support a new function for ALOXE3, i.e., an enzyme essential for the lipid mediator class switch that occurs during initial inflammatory and terminal resolution stages of the vascular response to injury.
Limits of the research and proposed minor revisions:
- The paper will benefit from a cartoon schematically showing the possible signaling pathways which triggers the activation of APOX3 in the endothelial cells and comparison with the pathways that regulate APOX15B.
Response: We appreciate the positive comments of the reviewer. Given the license to speculate in the next comment, we have taken advantage of this proposal to make a carton to postulate which cellular processes mediate ALOX expression and devised a new figure (Figure 7) that couples to the paragraph added to the Discussion (see below).
- Please add one extra paragraph in the discussion in which you contrast the signal transduction pathways leading to the activation/inhibition of both APOX15B and APOX3 in the endothelial cells under healthy and pathophysiological conditions (like damaged endothelium!).
Response: A new paragraph has been added to the Discussion as suggested by the reviewer. This paragraph attempts to link together a variety of recent data that provide a new understanding regarding the early events surrounding the transition from the healthy state to repair of vascular damage, and how the signal transduction systems operating under these conditions might influence ALOX function. We have tried to incorporate the concepts illustrated in the graphical abstract so that all aspects of the manuscript will linked together.